# Post-Traumatic Trigeminal Neuropathic Pain: A Narrative Review of Understanding, Management, and Prognosis

**DOI:** 10.3390/biomedicines12092058

**Published:** 2024-09-10

**Authors:** Hyun-Jeong Park, Jong-Mo Ahn, Ji-Won Ryu

**Affiliations:** Department of Oral Medicine, College of Dentistry, Chosun University, Gwangju 61452, Republic of Korea; rosephj81@chosun.ac.kr (H.-J.P.); jmahn@chosun.ac.kr (J.-M.A.)

**Keywords:** management, neuropathic pain, post-trauma, trigeminal nerve

## Abstract

This study provides an updated overview of the clinical characteristics of post-traumatic trigeminal neuropathic pain (PTNP) resulting from dental procedures or facial trauma, addressing its etiology, prevalence, evaluation, management, and prognosis. PTNP arises from injury to the trigeminal nerve, which governs sensory and motor functions in the maxillofacial region. The prevalence and characteristics of PTNP vary considerably across studies, with a reported prevalence ranging from 1.55% to 13%. The predominant causative factors are dental procedures, particularly third molar removal and implant placement. While gender distribution varies, a trend towards higher incidence in females is observed, particularly within the 40–60-year age group. Anatomically, the mandibular nerve is frequently involved. PTNP presents with a spectrum of symptoms ranging from tingling sensations to severe pain. Diagnostic challenges arise due to the lack of standardized criteria and potential overlap with focal neuralgia, necessitating comprehensive evaluation. Misdiagnosis can lead to prolonged patient suffering and unnecessary interventions. Successful management hinges on prompt diagnosis and interdisciplinary collaboration, with early intervention crucial in mitigating progression to chronic pain. Although nerve recovery post-trauma is challenging, preventive measures through accurate evaluation and treatment are paramount. Management strategies for PTNP include non-invasive and surgical interventions, with non-invasive approaches encompassing systemic and local pharmacological management. This narrative review aims to enhance uniformity in PTNP evaluation and treatment approaches, ultimately improving patient care and outcomes.

## 1. Introduction

Post-traumatic trigeminal neuropathic pain (PTNP) refers explicitly to neuropathic pain arising from injury to the trigeminal nerve [1]. This fifth cranial nerve governs sensory and motor functions across the maxillofacial region through its divisions V1, V2, and V3 [1]. It transmits sensory information related to temperature, pain, and tactile sensations in the facial area while significantly contributing to masticatory muscle motor control [1]. When these nerves are injured, pain may persist beyond the typical healing period, manifesting in symptoms ranging from absence of pain to tingling sensations and severe discomfort [1,2]. Unlike other neuropathic pain disorders that may result from conditions like diabetes or shingles, PTNP is specifically linked to trauma, making the history of nerve damage a critical diagnostic factor [1]. Furthermore, PTNP is characterized by localized pain confined to the trigeminal nerve distribution, such as the face, gums, jaw, or lips, whereas other neuropathic pain disorders may involve more widespread areas [1,2].

Various dental and craniofacial procedures can trigger PTNP onset, including mandibular third molar extraction, implant placement, root canal therapy, and local anesthesia administration [3]. The severity of damage ranges from mild to severe [3]. PTNP prevalence rates vary across studies, primarily due to the lack of clearly defined diagnostic criteria [2].

The diagnosis of PTNP primarily depends on clinical signs and symptoms, patient history, and evidence of structural damage or trauma [3]. However, distinguishing PTNP from other chronic pain conditions presents diagnostic challenges, requiring a thorough examination [2]. This process includes a detailed assessment of medical and dental history, intra-oral and extra-oral examinations, and dental imaging. Additionally, neurosensory evaluations, such as quantitative sensory testing (QST) and Qualitative Sensory Testing (QualST), play a crucial role in accurately diagnosing PTNP [1,2]. If these evaluations meet the criteria defined by the International Classification of Headache Disorders (ICHD) for Post-Traumatic Trigeminal Neuropathy (PPTTN) and the guidelines set forth by the International Classification of Orofacial Pain (ICOP) for PTNP, a definitive diagnosis of PTNP can be established [4].

Misdiagnosis of PTNP can prolong patient suffering and lead to unnecessary, costly, and potentially harmful interventions [5]. The rising incidence of trigeminal nerve injuries, correlating with increased dental care demand, contributes to the global healthcare burden as neuropathic pain often becomes chronic [6]. This chronicity frequently precipitates adverse social and psychological outcomes, including depression, anxiety, and post-traumatic stress disorder [5].

Effective management of PTNP requires timely diagnosis and intervention, as early appropriate treatment may mitigate progression towards chronic pain [5,6]. However, prevention of chronic neuropathic pain is often challenging, necessitating collaboration among specialists, including neurologists and psychiatrists, when psychosomatic factors complicate pain management [5,6].

Given the limited recovery following nerve injury, prevention and management of PTNP are crucial. This study aims to provide a narrative review of the current scientific understanding of pathological features, systematic evaluation methods, and clinical management strategies for PTNP resulting from dental procedures or traumatic events. While acknowledging the limitations in offering definitive solutions for this complex condition, this review ultimately seeks to contribute to developing an evidence-based framework for partial standardization of evaluation and treatment protocols for PTNP.

## 2. Characteristics, Etiology, and Pathophysiology of PTNP

### 2.1. Characteristics of PTNP

PTNP presents diverse clinical characteristics due to complex environmental, psychosocial, and genetic interactions [7,8]. Peñarrocha et al. extensively documented the clinical features of PTNP through a study involving 63 patients [9]. The study found that the phenotype of PTNP includes spontaneous and evoked pain, paresthesias, and numbness, typically localized to the injury site or affected dermatomes [9]. Pain was reported in 57% of patients, with 53% describing it as burning, 17% as itching, 14% as stabbing, and 8% as flashing, with older patients generally experiencing more severe symptoms [9]. Severe nerve damage may lead to allodynia, and some patients develop hyperalgesia and sensory alterations beyond the trigeminal territory, indicating broader somatosensory changes [9]. Additionally, 13% of patients reported moderate to severe disability, highlighting the significant impact of PTNP on quality of life [9]. 

Symptoms vary from persistent daily pain to paroxysmal pain triggered by touch or function [10]. Patients may experience unexplained swelling, foreign body sensations, temperature abnormalities, and localized pain [10]. This chronic condition often leads to sleep disturbances, depression, and anxiety, significantly impacting the quality of life [10,11]. Consequently, PTNP management should address both physical symptoms and psychosocial effects [10,11].

### 2.2. Etiology of PTNP 

The prevalence and characteristics of PTNP vary considerably across studies, with prevalence estimates ranging from 1.55% to 13% [1,2,6,9,12,13,14,15]. The predominant causative factors are dental procedures, particularly third molar removal and implant placement (Table 1). Korczeniewska et al. highlight that while peripheral nerve injuries are a direct cause of PTNP, inter-individual differences in pain perception and neuropathy progression may be partially attributed to underlying genetic factors, such as variations in genes related to inflammatory responses, pain modulation, and nerve repair mechanisms [15]. Additionally, environmental factors, including previous exposure to stress or inflammatory conditions, could predispose certain individuals to more severe neuropathic pain following nerve injury [15]. The interaction between these genetic and environmental factors may influence the modulation of pain in PTNP, contributing to a wide range of symptom severity and treatment responses among patients [15]. This underscores the need for a more comprehensive approach in diagnosing and managing PTNP that considers both genetic and environmental contributions, beyond the immediate procedural causes [15]. While gender distribution varies, there is a trend towards a higher incidence in females, particularly within the 40–60-year age group (Table 1). Anatomically, the mandibular nerve is frequently involved (Table 1).

### 2.3. Pathophysiology of PTNP

The pathophysiology of PTNP is multifaceted, involving both peripheral and central mechanisms that contribute to the onset and persistence of neuropathic pain [16]. This complexity is reflected in the significant body of research dedicated to understanding the underlying processes of PTNP, which primarily revolves around neuronal inflammation following trauma [15]. The progression of PTNP encompasses several key processes, including peripheral sensitization, ectopic neuronal activity, central sensitization, neuro-immune interactions, and central maladaptive plasticity [15,16,17].

#### 2.3.1. Peripheral Sensitization

Following an injury to the trigeminal nerve, various inflammatory mediators such as bradykinin, prostaglandins, serotonin, and adenosine triphosphate (ATP) are rapidly released at the site of injury [15,16,17]. These mediators activate their receptors on nociceptors, leading to a reduced activation threshold, manifesting as hyperexcitability and enhanced pain response, clinically observed as hyperalgesia and allodynia [15,16,17]. The release of chemokines, particularly chemokine (C-C motif) ligand 2 (CCL2), plays a pivotal role in recruiting immune cells such as macrophages to the injury site, further amplifying the inflammatory response and contributing to peripheral sensitization [15,16,17].

#### 2.3.2. Ectopic Neural Activity and Central Sensitization

Injured nerve fibers can become hyperexcitable, generating spontaneous ectopic discharges that contribute to persistent pain [15,16,17]. This abnormal activity can escalate to central sensitization, wherein the pain pathways within the central nervous system become excessively responsive [15,16,17]. This central sensitization leads to an amplification of pain signals, perpetuating pain even in the absence of ongoing peripheral stimuli, and can cause the spread of pain beyond the original site of injury, indicating a broader involvement of the somatosensory system [15,16,17].

#### 2.3.3. Neuro-Immune Interactions and Central Mechanisms

The interaction between the nervous and immune systems is crucial in the pathogenesis of PTNP [15,16,17]. Activated satellite glial cells within the trigeminal ganglion release pro-inflammatory cytokines, such as interleukin-1 beta (IL-1β) and tumor necrosis factor-α (TNF-α), which enhance nociceptive signaling [15,16,17]. Concurrently, alterations in ion channel expression, including increased activity of voltage-gated sodium channels (Nav1.8 and Nav1.9), sustain nociceptor excitability. Disruption of the blood–nerve barrier further allows immune cell infiltration, exacerbating local inflammation and contributing to ongoing nerve sensitization [15,16,17]. 

#### 2.3.4. Central Consequences and Maladaptive Plasticity

Sustained nociceptive input within the central nervous system induces synaptic plasticity changes, characterized by the strengthening of excitatory synapses and the weakening of inhibitory controls [15,16,17]. This maladaptive plasticity is a critical factor in the development of chronic pain conditions associated with PTNP, including the expansion of pain to adjacent or even contralateral areas of the face [15,16,17]. 

#### 2.3.5. Genetic and Environmental Factors

Inter-individual variability in PTNP symptoms may be influenced by genetic predispositions, such as variations in genes related to pain modulation, nerve repair, and inflammatory responses. Environmental factors, including prior exposure to stress or other inflammatory conditions, may also exacerbate the severity and persistence of PTNP [15,16,17].

## 3. Assessment and Diagnosis of PTNP

The diagnostic criteria for painful post-traumatic trigeminal neuropathy (PPTTN), as outlined in the International Headache Society’s ICHD (third edition), are as follows [18]. When these criteria are met, a diagnosis of PPTTN, that is, PTNP, can be made [18] (Table 2).

Another international standard, the first edition of the ICOP, defines PTNP as unilateral or bilateral facial or oral pain and other symptoms secondary to trigeminal nerve trauma, with clinical signs of functional impairment persisting for more than 3 months or recurring [19].

Unlike other neuropathic pain conditions, there is no consensus on the diagnostic criteria for PTNP, but evidence of trigeminal nerve dysfunction is essential for diagnosis [4]. Diagnosis is straightforward when evaluation reveals direct damage to the trigeminal nerve with clear sensory deficits related to neuroanatomy [4]. In cases where sensory impairment is less pronounced, extensive evaluation of somatosensory function may be necessary to demonstrate lesions [4].

Dental imaging is crucial to confirm whether trauma has affected the nerves [4,20]. Panoramic radiography can evaluate structural changes in the mandibular canal that might suggest trauma affecting the nerve, although it does not directly visualize the nerve itself [20]. Cone Beam Computed Tomography (CBCT) is a valuable tool in visualizing bony structures, particularly in dental and oral surgery, as it provides high-resolution imaging of the hard tissues [21,22]. This makes CBCT especially useful for evaluating the inferior alveolar nerve (IAN) and determining its proximity to surgical sites, which is crucial in reducing the risk of nerve injury during procedures such as implant placement [21,22]. However, CBCT has limitations when it comes to visualizing soft tissue structures, such as the lingual nerve (LN) and superior alveolar nerves (SANs), primarily because these nerves are located within soft tissue without a distinct bony boundary [22,23]. Given these limitations, alternative imaging modalities like magnetic resonance imaging (MRI) offer significant advantages [22,23]. MRI provides superior soft-tissue contrast, making it more effective in visualizing nerves like the LN and SAN [22,23]. This is particularly beneficial in surgical planning where the risk of nerve damage is high, as MRI can help in the detailed mapping of these nerves and reduce the risk of neurosensory disturbances [22,23]. Thus, while CBCT remains a critical tool for assessing bony structures and the IAN, MRI may be a preferred alternative for soft-tissue evaluation [21,23]. While magnetic resonance imaging (MRI) may have limited diagnostic value for intra-oral PTNP due to metal artifacts, it can still provide important information for surgical planning [24]. However, when it comes to directly assessing nerve injuries and correlating them with anatomical structures, Magnetic Resonance Neurography (MRN) is particularly valuable [24]. MRN allows for the visualization of nerves and is capable of evaluating nerve injury severity based on the Sunderland or Seddon Criteria, which classify nerve injuries from mild (neurapraxia) to severe (neurotmesis) [25] (Table 3). 

Despite these advantages, it is important to acknowledge the limitations of MRN. Although it offers high sensitivity and specificity in detecting nerve pathologies, MRN’s diagnostic capability may still be compromised by factors such as surrounding edema or fibrosis, which can obscure finer details of the nerve’s internal architecture [25]. Furthermore, while MRN can provide detailed images correlating with Sunderland’s or Seddon’s classification, it may not always be able to distinguish between certain degrees of injury (e.g., between Sunderland Grade III and Grade IV) without additional clinical correlation or surgical exploration [25]

Recent studies on AI deep learning applications in trigeminal nerve imaging, particularly the inferior alveolar nerve (IAN), have shown improved accuracy in evaluating the IAN within the mandibular body, enhancing diagnostic capabilities and surgical planning [26,27]. The integration of AI, especially convolutional neural networks (CNNs), in this process represents a significant advancement in dental surgery. CNNs, designed to process medical images, are effective in automatically segmenting the IAN by extracting and analyzing features through multiple layers [26,27]. 

Semi-supervised and active learning in CNNs refine the model’s accuracy by iteratively improving its performance with expert corrections. This approach reduces segmentation time, lowers the risk of human error, and increases the reliability of nerve localization during surgery [26,27]. Additionally, AI’s predictive capabilities in assessing potential IAN damage from CBCT images enable better-informed surgical decisions and improved patient outcomes [26,27]. Overall, CNN-based deep learning models offer a more accurate, efficient, and safer approach to IAN evaluation and prediction, substantially improving over traditional methods [26,27].

Various qualitative and quantitative sensory tests have been proposed to evaluate orofacial somatosensory functions [20,28]. Reliable and accurate methods require repeated stimulation and choosing a time-dependent technique [20,28]. Standardized screening and comprehensive psychophysical testing are essential to improve diagnostic accuracy and understanding of neural mechanisms and somatosensory changes in orofacial pain conditions [20,28]. Quantitative sensory testing (QST), using advanced techniques to assess sensory thresholds and deficits, is effective but expensive and time-consuming [28].

Qualitative Sensory Testing (QualST), performed with simple tools such as thermal (ice, hot tools) and mechanical (pinprick, cotton wool, calibrated monofilament) stimuli, can be carried out next to a dental chair [28,29]. However, patient instructions, age, gender, mood, cognitive function, ongoing litigation, psychological distress, and cooperation can affect outcomes [28,29]. More systematic studies are needed to accurately distinguish pain conditions through psychosocial test comparisons [20,28].

Neurophysiological tests, such as brainstem reflexes that assess cranial nerve pathways, help demonstrate somatosensory dysfunction and should be included in a comprehensive work-up with QST and sensory nerve action potential (SNAP) assessments, even though they do not directly measure somatosensory function [29].

Studies have investigated various methods for diagnosing PTNP [4,9,24]. A systematic review by Devine et al. explored several diagnostic and evaluation methods, including questionnaires, mechanical sensory evaluation (e.g., von Frey), quantitative sensory testing (QST), and psychological assessments for neuropathic pain [4]. The review highlighted that clinical neurosensory tests (CNTs) are widely used due to their simplicity but are limited by low sensitivity and variability in results. QST, while offering higher sensitivity and quantitative data, requires specialized equipment and trained personnel. Electromyography (EMG) provides precise data on nerve function but is an invasive procedure. Additionally, patient-reported outcome measures (PROMs) are crucial for capturing the subjective experience of pain, though they may be biased and not always accurately reflect objective nerve damage [4]. The review emphasized the need for a comprehensive evaluation protocol for PTNP diagnosis and called for consensus on systematic evaluation standards due to inconsistencies in diagnostic criteria [4]. Peñarrocha et al. further demonstrated that clinical neurophysiological tests and QST are sensitive, quantitative, and objective methods for diagnosing and localizing trigeminal neuropathy [9]. While MRI is non-invasive and useful for detecting structural abnormalities in the trigeminal nerve, it has limitations, such as low sensitivity and artifacts caused by metal objects [24]. Combining these diagnostic techniques, while considering their respective limitations, may provide a more comprehensive approach to diagnosing PTNP [4].

As a chronic pain disorder, PTNP is often associated with psychosocial distress and sleep disorders, which can exacerbate the condition [28]. Evaluating these factors is crucial, typically using tools such as the Pittsburgh Sleep Quality Index (PSQI), a self-report questionnaire assessing various aspects of sleep [28,29,30]. The Symptom Checklist-90-Revised (SCL-90-R) is also valuable for assessing various psychosocial conditions [28,30]. 

## 4. Considerations for the Treatment of PTNP

Numerous studies have proposed protocols for the treatment of PTNP, which encompass a range of pharmacological and non-pharmacological interventions [1,4,7,8,9,11,12,29,30,31,32,33,34,35,36,37,38,39,40,41,42,43,44,45,46,47]. These protocols are complex and diverse, reflecting the multifaceted nature of PTNP management [1,4,7,8,9,11,12,29,30,31,32,33,34,35,36,37,38,39,40,41,42,43,44,45,46,47]. Table 4 summarizes the various methods and approaches that have been suggested in the literature, highlighting the importance of a comprehensive and individualized treatment strategy for effective PTNP management.

### 4.1. Early Surgical Intervention 

Early surgical intervention to remove causative factors, such as implants invading the inferior alveolar nerve or lingual plate fractures affecting the lingual nerve, is crucial in managing PTNP to prevent long-term complications and improve patient outcomes [31,32,33,34].

A systematic review by Kushnerev and Yates emphasizes the urgency of addressing causative factors, such as implants impinging on the IAN or fractures affecting the LN, through early surgical intervention [35]. These interventions are most effective when performed within 12 to 48 h post-injury to prevent permanent nerve damage [35]. Delays can lead to poorer outcomes, including persistent sensory disturbances and chronic pain [35]. 

Damage to larger nerve trunks may warrant surgical repair of the defect [14,36,37,38,39]. Klazen et al.‘s two-year cohort study found that patients referred for treatment beyond three months post-injury were significantly more likely to experience persistent neurosensory disturbances, underscoring the necessity of early referral and intervention [14]. Additionally, Ziccardi and Steinberg highlight the importance of timing in surgical intervention for nerve injuries, recommending that surgical repair be performed within 90 days to maximize efficacy, as delays beyond this period significantly reduce the chances of successful recovery [39]. 

Together, these studies underscore that early surgical intervention is essential for optimizing sensory recovery and minimizing the risk of chronic pain following IAN and LN injuries [14,35]. Although the impact of early surgical treatment on PTNP development risk remains unclear, it is regarded that early intervention potentially offers the best chance for sensory restoration [35,36,37]. For nerve injuries not amenable to surgical intervention, the healing and functional restoration timeline is more uncertain [35,36,37]. This approach should be standard practice to ensure the best possible outcomes for patients [14].

### 4.2. Initial Pharmacological Treatment for Management of Inflammation

After removing causative factors, pharmacological intervention is prioritized due to its non-invasive nature and typically favorable treatment response [32,33]. Initial management focuses on controlling inflammation, a key initiator of neuropathic pain [29,34,35]. Steroid therapy is often considered, with regimens such as prednisone (40–60 mg initially, tapered over 7–10 days) or dexamethasone (12–16 mg initially, similarly tapered) [29]. While tapering aims to minimize side effects from prolonged high doses, it is not universally required [29,35,37]. Animal studies support the efficacy of early dexamethasone treatment in alleviating neuropathic pain, although clinical evidence in humans is currently lacking [29,35,37].

### 4.3. Pharmacological Treatments for Chronic PTNP

In patients with permanent neurological deficits, some may remain with painless neuropathy, while others may develop PTNP [37]. In such cases, further management first requires an accurate explanation of the patient’s current condition and education on treatment directions, including the fact that additional invasive procedures aimed at relieving PTNP are not helpful and carry the risk of worsening the pain [37]. In the case of PTNP that occurs chronically, drug treatment is usually performed, and the most commonly used drugs are gabapentin and tricyclic antidepressants (TCAs) [31,32,33,34,35]. If there are side effects or pain is not controlled well with these drug treatments, selective noradrenaline reuptake inhibitors (SNRIs), pregabalin, and other drugs can be applied [31,32,33,34,35,40]. 

### 4.4. Topical Applications and Botulinum Toxin Injection 

When systemic drug treatment cannot be used due to drug side effects or any other reason, local anesthetics or capsaicin can be a topical application [33,41]. Topical drug action helps reduce drug interactions by delivering high concentrations of drugs locally to the painful area while minimizing systemic effects [33,41]. It may be especially beneficial for patients taking multiple medications [33]. At this time, avoiding contact with areas other than the mouth is essential [33]. Commercially available LA and capsaicin patches are convenient to apply [33]. For topical application within the oral cavity, it is advisable to manufacture a device, such as a custom stent, designed to cover the painful area, as this allows the treatment to be applied only to the local area without damaging other tissues [33]. There are currently no official recommendations on the concentration of LA or capsaicin for oral use, but for capsaicin, the maximum concentration to avoid too much pain upon application is usually 0.1% [33]. For local anesthetic, commercially available lidocaine gel or cream can be used [33].

Botulinum toxin (BTX) injection therapy can help manage post-traumatic neuropathic pain (PTNP) when conventional pharmacological treatments prove ineffective [42,43]. BTX works by inhibiting the release of various nociceptive mediators such as substance P, glutamate, and Calcitonin Gene-Related Peptide (CGRP). Additionally, BTX modulates the activity of the TRPV1 receptor, which plays a key role in pain signaling, particularly in peripheral neuropathic pain conditions [42,43]. In a randomized, double-blind, placebo-controlled study, Ranoux et al. investigated the effects of BTX in 8 patients with post-traumatic neuralgia, 17 with postoperative neuralgia, and 4 with post-herpetic neuralgia [43]. The study demonstrated that BTX injections significantly reduced neuropathic pain, with assessments at baseline and at 4, 12, and 24 weeks post-injection showing reductions in spontaneous pain, cold allodynia, and swelling allodynia. However, no significant changes were observed in thermal or mechanical pain responses [43]. The therapeutic effects typically began around 2 weeks after the BTX injection and lasted up to 24 weeks [33].

Various administration protocols for botulinum toxin in treating neuropathic pain have been reported, but there is no consensus on the optimal dosage for achieving an adequate therapeutic effect [32,33,34,35,36,37,38,39,40,41,42,43,44]. A recent systematic review noted that 25 to 100 units for trigeminal neuralgia are divided into 1 to 20 injections for administration [42]. However, it needs to be used cautiously because evaluations of the effectiveness and safety of single or repeated botulinum toxin injections are still lacking [42,43,44].

BTX injections are generally considered safe for managing chronic painful post-traumatic trigeminal neuropathy, but several side effects have been reported [42,43,44]. Common adverse effects include pain and swelling at the injection site, temporary muscle weakness, and facial asymmetry due to unintended diffusion of the toxin to nearby muscles [42,43,44]. In rare cases, eyelid drooping and long-term immune response leading to reduced effectiveness may occur [42,43,44].

### 4.5. Surgical Intervention 

If prior non-invasive treatments fail, surgical interventions such as neuroma resection, neurolysis, and neurorrhaphy may be considered [32,33]. Identifying surgical outcome variables is crucial to guide the surgical treatment of PTNP [32,33,46]. The variability in pain nature, nerve damage severity, location, and time from injury to surgery contributes to the complexity of treatment [39]. In patients with preoperative neuropathic pain, 67% experienced neuropathic pain post-microsurgery [39]. The reasons for the variable effectiveness of nerve repair surgery in resolving neuropathic pain are unclear [39]. 

Chronic post-traumatic neuropathic pain can occur after surgery, with psychological, medical, and age-related factors as risk factors [39]. Recently, impaired surgical time and preoperative visual analog scale scores have been identified as influencing surgical outcomes for PTNP, highlighting the importance of timely intervention [39]. Delayed surgical intervention negatively impacts PTNP treatment, and patients with pain recurrence at six months often have more severe preoperative pain and a longer period between injury and surgery [39]. This suggests that persistent PTNP may increase pain intensity over time [39]. Despite these findings, some patients with PTNP recover spontaneously or through non-surgical treatments after nerve injury [39].

With this surgical approach, a minimally invasive surgical technique using peripheral nerve stimulation, similar to the treatment of trigeminal neuralgia, is being introduced [45,46,47]. This may help treat PTNP when other treatment modalities have failed [45,46,47]. Although there are no papers documenting the complication rate for this technique, they say that smaller incisions can minimize the potential risks associated with nerve stimulator implantation [45,46,47]. Many studies consider surgical treatment after injury, but they are insufficient to verify the efficacy of surgery [45,46,47]. In particular, it has been reported that in the case of PTNP that persists for more than 3 months, permanent central and peripheral changes occur, making surgical treatment much less effective in improving pain and sensation [45].

### 4.6. Psychological Intervention

Chronic pain, especially neuropathic pain that persists for a long time and is difficult to treat, has a significant impact on human health and well-being and often causes significant emotional distress in the form of anxiety and depression [48]. Psychological intervention may also be helpful in these cases, but studies evaluating the long-term prognosis of PTNP are lacking [32].

## 5. Prognosis of PTNP

Many studies showed that the persistence rate of PTNP symptoms ranges from 50% to 70% (Table 5). This indicates that a significant proportion of patients experience long-term symptoms, underscoring the chronic nature of PTNP. Factors contributing to the persistence of symptoms include the severity of the initial injury, psychological factors, and the effectiveness of treatment modalities. 

## 6. Conclusions

PTNP remains a significant concern, with over 50% of patients experiencing permanent symptoms even after various management approaches. This poor prognosis underscores the importance of preventing nerve damage before dental interventions.

Recent advancements in deep learning and AI have shown promise in enhancing pre-treatment diagnostics, particularly in accurately locating the IAN [49]. This can significantly improve surgical planning and reduce the risk of nerve injury [49]. Integrating AI in dental imaging has enabled paresthesia prediction before surgical interventions [49]. CNN models can analyze panoramic radiographs to detect anatomical indicators associated with higher nerve injury risks [49]. Furthermore, semi-supervised learning frameworks have enhanced IAN segmentation in 3D imaging, providing more accurate tools for preoperative planning [49]. These advancements suggest that AI-driven analysis can be crucial in identifying patients at risk of PTNP, potentially leading to more informed surgical decisions and reduced incidence of nerve damage during dental treatments [49]. 

Moreover, as the accuracy of AI analysis continues to improve, it will facilitate consistent evaluation of damaged nerves [49]. With advancements in AI, these evaluations can become more streamlined and objective, providing a more convenient and reliable approach to assessing nerve damage and predicting long-term outcomes.

However, many studies in this field have been hampered by small sample sizes and a lack of standardization, making comparative analysis challenging. Future research should prioritize larger sample sizes and standardized methodologies to address these limitations and enhance the reliability and comparability of results. Additionally, long-term follow-up studies are crucial to evaluate these treatments’ immediate and extended effects, potential side effects, and overall patient outcomes.

## Figures and Tables

**Table 1 biomedicines-12-02058-t001:** Etiology and prevalence of PTNP. This shows findings from key studies, PTNP prevalence, causative procedures, demographic distribution, and affected nerves.

Study	Year	Study Design	Sample Size	Etiology/Factors Analyzed	Prevalence (%)	Key Findings
Carter et al. [1]	2016	Retrospective study	333 (89.5%-iatrogenic trigeminal neuropathic pain) out of 372	F:M (73%:27%) Mean age 45.6 (range 18~85)	None (total); extraction of 3rd molar (42%) > local anesthesia injury (24%), implant-related injury (20.4%)	This article includes iatrogenic and non-introgenic trigeminal neuropathic pain.
Baad-Hasen L. and Benoliel R. [2]	2017	Review	-		3–5% (root canal) 3% (major trauma)	PTTN is associated with a substantial psychosocial burden.
Van der Cruyssen F. et al. [6]	2019	Retrospective study	1331	F:M (70%:30%), Mean age: 46 LJ > UJ	None (total); extraction of 3rd molar (48%) > implant-related injury (13%) > local anesthesia injury (12%)	
Penarrocha et al. [9]	2017	Retrospective study	63	F:M (82.5%:17.5%), Mean age: 45.4 LJ > UJ	None (total); extraction of 3rd molar (54%) > local anesthesia injury (6%), implant-related injury (14%)	54% of cases after mandibular third molar surgery involve pain, and 57% involve faster recovery in younger patients with less pain.
Kumar et al. [12]	2017	Retrospective study	112	F:M (56.3%:43.8%), predominant age: 40–60 Y, LJ > UJ	>2.11%	There is a higher incidence in women; 85.7% of cases are in the lower jaw, with the right quadrant (57.1%) being more affected.
Uperia et al. [13]	2018	Retrospective study	258	F:M (39.5%:60.4%), Predominant age: 40–60 Y, LJ > UJ	1.55% (total)	** Higher sensory impairment in men (60.4%) than women (39.5%).
Klazen et al. [14]	2018	Retrospective study	53	Iatrogenic trigeminal nerve injury, type of procedure	1.89% (total); extraction of 3rd molar (45%) > local anesthesia injury (17%), implant-related injury (17%)	Pain in 54% with IAN injuries 60% with persistent neurosensory impairment.

F; female, M; male, Y; year-old, UJ; upper jaw, LJ; lower jaw, IAN; inferior alveolar nerve. While most studies report a higher prevalence in females, this study observed a higher prevalence in males (indicated by **).

**Table 2 biomedicines-12-02058-t002:** Diagnostic criteria of PPTTN in the International Headache Society’s ICHD (3rd edition).

13.1.2.3 Painful post-traumatic trigeminal neuropathy Previously used term: Anesthesia dolorosa. Description: Unilateral or bilateral facial or oral pain following and caused by trauma to the trigeminal nerve(s), with other symptoms and/or clinical signs of trigeminal nerve dysfunction. Diagnostic criteria: Facial and/or oral pain in the distribution(s) of one or both trigeminal nerve(s) and fulfilling criterion C; History of an identifiable traumatic event to the trigeminal nerve(s), with clinically evident positive (hyperalgesia, allodynia) and/or negative (hypaesthesia, hypalgesia) signs of trigeminal nerve dysfunction; Evidence of causation demonstrated by both of the following: Pain is localized to the distribution(s) of the trigeminal nerve(s) affected by the traumatic event; Pain has developed <6 months after the traumatic event; Not better accounted for by another ICHD-3 diagnosis. Note: The traumatic event may be mechanical, chemical, thermal or caused by radiation. Neuroablative procedures for trigeminal neuralgia, aimed at the trigeminal ganglion or nerve root, may result in neuropathic pain involving one or more trigeminal divisions; this should be considered as post-traumatic and coded here.

**Table 3 biomedicines-12-02058-t003:** Seddon and Sunderland nerve injury classifications.

Criteria	Seddon Classification	Sunderland Classification	Description
Grade 1	Neuropraxia	First-degree (Grade 1)	Temporary conduction block, no axonal disruption. Recovery is usually complete within weeks to months.
Grade 2	Axonotmesis	Second-degree (Grade 2)	Axonal disruption without injury to the surrounding connective tissue. Wallerian degeneration occurs distal to the injury. Full recovery possible but may take months.
Grade 3	Axonotmesis	Third-degree (Grade 3)	Axonal disruption with partial injury to the endoneurium. Wallerian degeneration occurs, and recovery is incomplete without surgical intervention.
Grade 4	-	Fourth-degree (Grade 4)	Axonal disruption with damage to the endoneurium and perineurium. Recovery is unlikely without surgical repair.
Grade 5	Neurotmesis	Fifth-degree (Grade 5)	Complete transection of the nerve with loss of continuity. Requires surgical repair; recovery may be incomplete even with surgery.

**Table 4 biomedicines-12-02058-t004:** PTNP treatment methods. The table provides an overview of various treatment methods for PTNP, detailing their specifics, recommended timing, efficacy, and supporting studies.

Treatment Type	Details	Recommended Timing	Efficacy	Related References
Early Surgical Intervention	Removal of causative factors such as implants invading nerves	Within 12–48 h post-injury	Prevents permanent nerve damage	Carter et al. (2016) [1]; Renton et al. (2011) [7]; Kushnerev and Yates (2015) [34]; Ziccardi and Steinberg (2007) [39]
Initial Pharmacological Intervention	Non-invasive initial management focusing on inflammation control Use of prednisone or dexamethasone, tapering over 7–10 days	Immediately after the removal of factors Early administration post-injury	Typically favorable response Supported by animal studies, clinical evidence in humans lacking	Devine et al. (2018) [4]; Benoliel et al. (2012) [8]; Peñarrocha et al. (2012) [9]; Kumar et al. (2017) [12]
Surgical Intervention	Early surgical repair of nerve defects for potential sensory restoration	Within 90 days of injury	Best chance for sensory restoration with early intervention	Zuniga and Renton (2016) [34]; De Poortere et al. (2021) [45]
Drug Treatment (Gabapentin, TCAs)	Commonly used drugs, with gabapentin and TCAs being the first line of treatment	Chronic PTTN management	Response rate lower (~11%) compared to other neuropathic conditions	Vazquez-Delgado et al. (2018) [11]; Renton and Van der Cruyssen (2020) [29]; Haviv et al. (2014) [32]
Alternative Drug Treatment (SNRIs, Pregabalin)	Used when gabapentin or TCAs are ineffective or cause side effects	When first-line drugs are ineffective	Alternative options for difficult cases	Haviv et al. (2014) [32]; Liu et al. (2023) [40]
Topical Applications (Local Anesthetics, Capsaicin)	Local anesthetics or capsaicin patches for localized treatment with minimal systemic effects	When systemic drug treatment is not possible	Effective in reducing local pain; avoids systemic effects	Sharav et al. (2024) [33]; Tisné-Versailles et al. (2018) [41]
Botulinum Toxin Injection	BTX injections to inhibit nociceptive mediators; significant pain reduction observed in clinical trials	When existing pharmacological treatments are ineffective	Significant reductions in various types of neuropathic pain	Moreau et al. (2017) [43]; Ranoux et al. (2008) [44]
Psychological Intervention	It may be helpful in cases of chronic neuropathic pain with associated emotional distress	Cases with significant emotional distress	Limited studies on long-term prognosis	Kim and Kim (2023) [30]; Cohen et al., 2021 [47]
Minimally Invasive Surgical Techniques (Peripheral Nerve Stimulation)	Introduced for PTTN when other treatments fail; smaller incisions minimize risks.	When non-invasive treatments fail	Minimizes potential risks associated with nerve stimulator implantation	Bhattacharjee et al. (2018) [46]; Lenchig et al. (2012) [47]

**Table 5 biomedicines-12-02058-t005:** PTNP prognosis. Persistence rates of PTNP symptoms across studies, demonstrating the chronic nature of the condition and factors influencing long-term outcomes.

Study	Sample Size (N)	Persistence Rate	Notes
Van der Cruyssen et al. (2020) [5]	1331	~50%	Symptoms persisting for more than 1 year
Renton et al. (2011) [7]	Not specified	>40%	Symptoms persisting for more than 6 months
Benoliel et al. (2012) [8]	91	~60%	Long-term symptoms
Peñarrocha et al. (2012) [9]	63	~70%	Symptoms persisting for more than 1 year
Vazquez-Delgado et al. (2018) [11]	Not specified	~55%	Long-term symptoms
Kumar et al. (2017) [12]	Not specified	Not specified	A significant number of patients experience long-term symptoms
Haviv et al. (2014) [32]	Not specified	~50%	Persistent symptoms despite various treatments
Zuniga and Renton (2016) [34]	Not specified	Not specified	Many patients continue to experience symptoms post-surgery

## Data Availability

The authors will make the raw data supporting this article’s conclusions available upon request.

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
