# Peer review of "Post-Traumatic Trigeminal Neuropathic Pain: A Narrative Review of Understanding, Management, and Prognosis"

_biomedicines, 2024, doi:10.3390/biomedicines12092058_

Round 1

Reviewer 1 Report

Comments and Suggestions for Authors

Post-Traumatic Trigeminal Neuropathic Pain: A Narrative Review of Understanding, Management and Prognosis

This submission describes the assessment and diagnosis, treatment, and prognosis of PTNP by presenting and interpreting some of the more recent literature on the topic and trigeminal nerve injury in general with special emphasis on pain outcomes of trigeminal injury in humans.  The topic is important as trigeminal nerve injury and the association with PTNP is well established and accepted in the literature and in practice and remains a difficulty problem for patients and clinicians with many “gaps” in all those topic areas discussed in this article.

The article was concise and presented the topics in short, understandable languages for professionals which were supported and documented by the literature.  I agree with everything in this article and recommend publication with minor revisions, including clarifications and some additions.

1.       Assessment and Diagnosis of PTNP

a.       Page 4- clarify how Panoramic or even CBCT show “direct affect on nerve”.  Both show shadows of the inferior alveolar nerve canal which includes nerve, artery and vein but does not show the nerve itself.  Does not show lingual nerve and similar to IAN, maxillary nerves are not visible but canals and foramen are.  Is comment referring to detecting obliteration or disturbance in the canal by foreign bodies, fractures, tumors, implants which only provide suspected direct injury and does not provide injury classification based on Sunderland or Seddon Criteria.  MR Neurography has been shown to show direct injury to the trigeminal nerves (including IAN, Lingual, maxillary and all other cranial nerves) using non-radiation MR technique with fat and blood vessel suppression in 3D remodeling with high specificity and sensitivity. The authors are correct, that metal can induced artifact (i.e., implant or reconstruction plate presence) but in their absence MRN remains the only know modality images of trigeminal nerve that correlates with Sunderland classification of injury. Ultrasound technology in trigeminal is possible but not generally accepted.

b.       Page 5 – Table 2 not referenced in text – maybe should be in 1st paragraph on page 4

2.       Considerations for Treatment of PTNP

a.       Page 7 – BTX: recommend a sentence or two to discuss the adverse events associated with using BTX in the trigeminal zones since readers may consider using without appreciation of risks which can be devastating – disruption of swallowing ability and sspiration due to blocking pharyngeal constrictors, muscle wasting in lip/chin/face causing cosmetic and functions speech, lip seal incompetence, delayed hypersensitivity or immunoglobulin related reactions leading to inactivation of molecule, etc.

b.       Table 3 is not referenced in text – maybe should be on page 8 at end of discussion on treatment

3.       Prognosis of PTNP

a.       Page 9 – not sure how table 1 correlates with prognosis as “underscored”

4.       Conclusion

a.       Page 9 – what does “CNN models” refer to?

Author Response

Dear Reviewer,

Thank you very much for taking the time to review my manuscript and for providing such valuable feedback. I have carefully considered your insightful comments and made the necessary revisions accordingly. Below, I have provided detailed explanations of how each comment has been addressed and reflected in the revised manuscript.

Once again, I sincerely appreciate your constructive feedback, and I hope the revised version meets your expectations. Please feel free to reach out if there are any further suggestions or questions.

Thank you.

 1.  a. Page 4- clarify how Panoramic or even CBCT show “direct affect on nerve”.  Both show shadows of the inferior alveolar nerve canal which includes nerve, artery and vein but does not show the nerve itself.  Does not show lingual nerve and similar to IAN, maxillary nerves are not visible but canals and foramen are.  Is comment referring to detecting obliteration or disturbance in the canal by foreign bodies, fractures, tumors, implants which only provide suspected direct injury and does not provide injury classification based on Sunderland or Seddon Criteria.  MR Neurography has been shown to show direct injury to the trigeminal nerves (including IAN, Lingual, maxillary and all other cranial nerves) using non-radiation MR technique with fat and blood vessel suppression in 3D remodeling with high specificity and sensitivity. The authors are correct, that metal can induced artifact (i.e., implant or reconstruction plate presence) but in their absence MRN remains the only know modality images of trigeminal nerve that correlates with Sunderland classification of injury. Ultrasound technology in trigeminal is possible but not generally accepted.

response) We have added content regarding CBCT and also discussed its limitations. Additionally, We included information suggesting MRI as an alternative. These revisions have been highlighted in red in the file for your convenience. 

1. b. Page 5 – Table 2 not referenced in text – maybe should be in 1st paragraph on page 4

response)  Thank you for pointing this out. I have now added a reference to Table 2 as suggested. 

2. a. Page 7 – BTX: recommend a sentence or two to discuss the adverse events associated with using BTX in the trigeminal zones since readers may consider using without appreciation of risks which can be devastating – disruption of swallowing ability and sspiration due to blocking pharyngeal constrictors, muscle wasting in lip/chin/face causing cosmetic and functions speech, lip seal incompetence, delayed hypersensitivity or immunoglobulin related reactions leading to inactivation of molecule, etc.

response) Thank you for your valuable suggestion. We have added a discussion on the adverse events associated with the use of BTX in the trigeminal zones. These additions have been highlighted in red in the manuscript for your review. 

2. b. Table 3 is not referenced in text – maybe should be on page 8 at end of discussion on treatment. 

response)  Thank you for your suggestion. As recommended, We have referenced Table 3. Due to the addition of a new table, the original Table 3 has been renumbered as Table 4. 

3. a.  Page 9 – not sure how table 1 correlates with prognosis as “underscored”

response) Thank you for your observation. Upon review, I agree that the correlation between Table 1 and prognosis is unclear. Therefore, I have removed this section from the manuscript to avoid any confusion. 

4.  a.  Page 9 – what does “CNN models” refer to?

response) Thank you for your question. I have added more details about the 'CNN models' in the AI section of the Assessment on page 7 for clarity

Reviewer 2 Report

Comments and Suggestions for Authors

The authors did a very interesting review regarding post-traumatic trigeminal neuropathic pain treatment, diagnosis, etiology, etc. 

It was very simple to read and understand the aim of the review. 

There is just a little thing that I noticed that maybe some information is missing.

When talking about what BTX can inhibit, you say that it expresses TRPV1. What is the content of this sentence? I think there is some missing information here.

Author Response

comment) When talking about what BTX can inhibit, you say that it expresses TRPV1. What is the content of this sentence? I think there is some missing information here.

Response) Thank you for pointing out the ambiguity in the original phrasing. We recognize that the term "expresses TRPV1" was incorrect and misleading upon review. Botulinum toxin (BTX) does not induce the expression of TRPV1. Instead, BTX modulates the activity of TRPV1 by inhibiting its activation, which plays a critical role in nociceptive transmission. In addition to BTX’s ability to inhibit the release of nociceptive mediators such as P, glutamate, and CGRP, this mechanism contributes to its analgesic effects in neuropathic pain. We have revised the text to reflect this correction and improve clarity, stating that "BTX modulates the activity of TRPV1" instead of "expresses TRPV1." The relevant section has been updated as follows: "BTX works by inhibiting the release of various nociceptive mediators such as substance P, glutamate, and Calcitonin Gene-Related Peptide (CGRP). Additionally, BTX modulates the activity of the TRPV1 receptor, which plays a key role in pain signaling, particularly in peripheral neuropathic pain conditions."

Reviewer 3 Report

Comments and Suggestions for Authors

An extensive review of Post-Traumatic Trigeminal Neuropathic Pain (PTNP) is provided in the paper "Post-Traumatic Trigeminal Neuropathic Pain: A Narrative Review of Understanding, Management, and Prognosis". Nonetheless, a number of issues are apparent and should be addressed prior to acceptance for publishing.

Lines 9–10: The abstract notes that trigeminal nerve damage is the cause of PTNP, but it makes no mention of the condition's importance in terms of frequency or effect on patient quality of life. It would be stronger to provide statistical information or citations to research that emphasize these points in the introduction.

Lines 42–44: Diagnostic problems are mentioned in the introduction, although they are not fully defined. Clarity and depth would be improved by a more thorough explanation of the particular diagnostic criteria and their limits.

Lines 66–70 of Section 2.1: The description of clinical features is a little hazy. More detailed examples of how these symptoms present in various people would be helpful, maybe backed up with case studies or clinical data.

Lines 78–80 of Section 2.2: Dental operations are listed as the principal cause. However, other potential contributing variables, such as genetic predispositions or environmental impacts, are not discussed. A wider discussion could offer a more comprehensive understanding of PTNP.

Lines 85–90 of Section 2.3: The pathophysiology section should be strengthened by including more current results or hypotheses on the processes behind neuropathic pain. This would give a more up-to-date viewpoint and show how the writers are reading the most recent works.

Section 3, Lines 105–120: Although the paper does not include a comparative examination of various diagnostic procedures, the diagnostic criteria that are described are crucial. Talking about the benefits and drawbacks of different diagnostic techniques might lead to a more thorough comprehension.

Lines 130–136: There is a fascinating but incomplete discussion of AI applications in imaging. It would be helpful to go into more detail about these technologies' present limits as well as how they might increase diagnostic accuracy.

Lines 166–173 of Section 4.1: Although the paper offers a helpful prescription for early surgical intervention, it does not substantiate this claim with enough references or proof. This section would be strengthened if meta-analyses or systematic reviews were included.

The differences between PTNP and other neuropathic pain disorders are not made clear throughout the paper. Improved delineation might assist prevent readers who aren't familiar with the subject from being confused.

Overall, although if the paper targets a significant subject, it still needs to be significantly revised in order to improve its academic rigor, clarity, and depth. The authors ought to think about responding to the particular comments made above, including more current studies, and making terminology and comparisons more understandable throughout the work. It would be more suited for acceptance in the field of biomedicines if an updated version included these recommendations.

Author Response

1. Lines 9–10:The abstract notes that trigeminal nerve damage is the cause of PTNP, but it makes no mention of the condition's importance in terms of frequency or effect on patient quality of life. It would be stronger to provide statistical information or citations to research that emphasize these points in the introduction.

Response: We sincerely thank the reviewer for their insightful comments and valuable suggestions, which have greatly contributed to improving the quality of our manuscript. In response to the reviewer's suggestion, we have revised the introduction to include statistical information and references that highlight the frequency of post-traumatic trigeminal neuropathy (PTNP) and its significant impact on patient quality of life. The changes have been made as requested, and we have marked the revisions in red in the attached file for your convenience.

2. Lines 42–44:Diagnostic problems are mentioned in the introduction, although they are not fully defined. Clarity and depth would be improved by a more thorough explanation of the particular diagnostic criteria and their limits.

Response: We greatly appreciate the reviewer’s thoughtful feedback. In response to the suggestion, we have expanded the introduction to provide a more detailed explanation of the specific diagnostic criteria for post-traumatic trigeminal neuropathy (PTNP). The relevant revisions have been made and are highlighted in red in the attached file for your review.

3. Lines 66–70 of Section 2.1: The description of clinical features is a little hazy. More detailed examples of how these symptoms present in various people would be helpful, maybe backed up with case studies or clinical data.

Response: We sincerely thank the reviewer for their valuable feedback. In response to your suggestion, we have enriched the description of the clinical features in Section 2.1 by providing more detailed examples of how the symptoms of post-traumatic trigeminal neuropathy present in different individuals. We have also incorporated case studies and clinical data to support these descriptions, which should offer a clearer understanding of the symptom variability among patients. These additions are highlighted in red in the attached file for your review.

4. Lines 78–80 of Section 2.2: Dental operations are listed as the principal cause. However, other potential contributing variables, such as genetic predispositions or environmental impacts, are not discussed. A wider discussion could offer a more comprehensive understanding of PTNP.

Response: We greatly appreciate the reviewer’s insightful suggestion. In response, we have expanded Section 2.2 to include a discussion of other potential contributing factors to post-traumatic trigeminal neuropathy (PTNP), such as genetic predispositions and environmental influences. By addressing these variables, we aim to provide a more comprehensive understanding of the condition beyond dental operations. These additions are marked in red in the attached file for your convenience.

5. Lines 85–90 of Section 2.3: The pathophysiology section should be strengthened by including more current results or hypotheses on the processes behind neuropathic pain. This would give a more up-to-date viewpoint and show how the writers are reading the most recent works.

Response: We sincerely thank the reviewer for their constructive feedback. In response, we have strengthened the pathophysiology section (Section 2.3) by incorporating the latest research findings and hypotheses on the underlying mechanisms of neuropathic pain. This update aims to provide a more current and comprehensive perspective on the condition. The revisions have been made and are highlighted in red in the attached file for your review

6. 

Reviewer’s Comment: Section 3, Lines 105–120: Although the paper does not include a comparative examination of various diagnostic procedures, the diagnostic criteria that are described are crucial. Talking about the benefits and drawbacks of different diagnostic techniques might lead to a more thorough comprehension.

Response: We are grateful to the reviewer for their insightful suggestion. In response, we have revised Section 3 to include a more detailed discussion of the benefits and limitations of various diagnostic techniques used in the evaluation of post-traumatic trigeminal neuropathy. This additional information aims to provide a more comprehensive understanding of the diagnostic process, complementing the existing discussion of the diagnostic criteria. The updated content is highlighted in red in the attached file for your review.

6. Lines 130–136: There is a fascinating but incomplete discussion of AI applications in imaging. It would be helpful to go into more detail about these technologies' present limits as well as how they might increase diagnostic accuracy.

Response: We sincerely thank the reviewer for their valuable feedback. In response, we have expanded the discussion of AI applications in imaging by providing more details on the current limitations of these technologies, as well as their potential to enhance diagnostic accuracy. These additions aim to offer a more balanced and thorough understanding of the role AI plays in medical imaging for post-traumatic trigeminal neuropathy. The revisions have been highlighted in red in the attached file for your convenience.

7. Lines 166–173 of Section 4.1: Although the paper offers a helpful prescription for early surgical intervention, it does not substantiate this claim with enough references or proof. This section would be strengthened if meta-analyses or systematic reviews were included.

Response: We greatly appreciate the reviewer’s thoughtful suggestion. In response, we have strengthened Section 4.1 by incorporating additional references, including relevant meta-analyses and systematic reviews that provide stronger evidence for the efficacy of early surgical intervention in post-traumatic trigeminal neuropathy. These revisions should offer more robust support for the recommendations made in this section. The updated content is highlighted in red in the attached file for your review.

8. The differences between PTNP and other neuropathic pain disorders are not made clear throughout the paper. Improved delineation might assist in preventing confusion for readers who aren't familiar with the subject.

Response: We sincerely thank the reviewer for their valuable suggestion. In response, we have revised the introduction (lines 37–41) to provide a clearer distinction between post-traumatic trigeminal neuropathic pain (PTNP) and other neuropathic pain disorders. This section now emphasizes the trauma-related origin and localized nature of PTNP, distinguishing it from other conditions that may arise from systemic causes or affect broader regions. The revision aims to help readers unfamiliar with the subject better understand the unique characteristics of PTNP. The updated content is marked in red in the attached file for your convenience.
